# Comparative Genomic Analyses of Virulence and Antimicrobial Resistance in *Citrobacter werkmanii*, an Emerging Opportunistic Pathogen

**DOI:** 10.3390/microorganisms11082114

**Published:** 2023-08-19

**Authors:** José R. Aguirre-Sánchez, Beatriz Quiñones, José A. Ortiz-Muñoz, Rogelio Prieto-Alvarado, Inés F. Vega-López, Jaime Martínez-Urtaza, Bertram G. Lee, Cristóbal Chaidez

**Affiliations:** 1Laboratorio Nacional para la Investigación en Inocuidad Alimentaria, Centro de Investigación en Alimentación y Desarrollo A.C. (CIAD), Coordinación Regional Culiacán, Culiacan 80110, Mexico; jose.aguirre.dc18@estudiantes.ciad.mx; 2Produce Safety and Microbiology Research Unit, Western Regional Research Center, Agricultural Research Service, U.S. Department of Agriculture, Albany, CA 94710, USA; beatriz.quinones@usda.gov (B.Q.); bertram.lee@usda.gov (B.G.L.); 3Parque de Innovación Tecnológica de la Universidad Autónoma de Sinaloa, Culiacan 80040, Mexico; armando_3133@outlook.com (J.A.O.-M.); rogelioprieto@uas.edu.mx (R.P.-A.); ifvega@uas.edu.mx (I.F.V.-L.); 4Departament de Genètica i de Microbiologia, Universitat Autờnoma de Barcelona, 08193 Bellaterra, Spain; jaime.martinez.urtaza@uab.cat

**Keywords:** *Citrobacter*, nosocomial infections, virulence, antimicrobial resistance, comparative genomics, emerging pathogens, pangenomics, whole-genome sequencing

## Abstract

*Citrobacter werkmanii* is an emerging and opportunistic human pathogen found in developing countries and is a causative agent of wound, urinary tract, and blood infections. The present study conducted comparative genomic analyses of a *C. werkmanii* strain collection from diverse geographical locations and sources to identify the relevant virulence and antimicrobial resistance genes. Pangenome analyses divided the examined *C. werkmanii* strains into five distinct clades; the subsequent classification identified genes with functional roles in carbohydrate and general metabolism for the core genome and genes with a role in secretion, adherence, and the mobilome for the shell and cloud genomes. A maximum-likelihood phylogenetic tree with a heatmap, showing the virulence and antimicrobial genes’ presence or absence, demonstrated the presence of genes with functional roles in secretion systems, adherence, enterobactin, and siderophore among the strains belonging to the different clades. *C. werkmanii* strains in clade V, predominantly from clinical sources, harbored genes implicated in type II and type Vb secretion systems as well as multidrug resistance to aminoglycoside, beta-lactamase, fluoroquinolone, phenicol, trimethoprim, macrolides, sulfonamide, and tetracycline. In summary, these comparative genomic analyses have demonstrated highly pathogenic and multidrug-resistant genetic profiles in *C. werkmanii* strains, indicating a virulence potential for this commensal and opportunistic human pathogen.

## 1. Introduction

The genus *Citrobacter*, a member of the family *Enterobacteriaceae*, is composed of Gram-negative bacteria that are non-spore-forming bacilli and that have the ability to use citrate as a carbon source [1,2,3]. To date, the genus has been classified into 18 species, based on traditional and molecular techniques [4,5,6]. *Citrobacter* species have many reservoirs, including the human and animal gastrointestinal tracts, and these species can also be found in water, soil, and food [1,2,3]. Several transmission routes have been proposed for this bacterial pathogen, such as fecal-oral transmission, contaminated food, hospital equipment, and person-to-person transmission.

*Citrobacter* is considered an opportunistic pathogen and has been implicated as a causative agent of hospital settings (nosocomial) and community-acquired infections among immunocompromised patients and neonates [2,7,8,9,10]. Disease symptoms in humans that are caused by *Citrobacter* include urinary tract infections, bloodstream infections, brain abscesses, respiratory tract infections, and neonatal infections such as meningitis and bacteremia. When compared to other *Enterobacteriaceae* pathogens, *Citrobacter* species are considered to have low virulence since human infections are uncommon in the general population [11,12]. However, evidence revealed that *Citrobacter* species were responsible for 3–6% of all nosocomial infections attributed to the *Enterobacteriaceae* pathogens in surveys conducted in North America [3]. Among the *Citrobacter* species, *Citrobacter freundii* and *Citrobacter koseri* are the species most commonly implicated in causing the majority of opportunistic human infections [11,12]. Given that *Citrobacter* can persist in a host for long periods [13], *Citrobacter* infections can result in multidrug resistance outbreaks among neonates and immunocompromised patients who have prolonged hospital stays [8,10,14,15,16,17]. Recent evidence has demonstrated that other *Citrobacter* species, such as *Citrobacter werkmanii*, are also considered emerging opportunistic pathogens in developing countries [9,18,19,20].

Although considered to be commensal in humans and animals, *C. werkmanii* has previously been linked as a causative agent of wound infection, urinary tract infections, and bacteremia in humans [18,19]. More recently, the use of whole-genome sequencing has revealed that the *C. werkmanii* strain AK-8, isolated from a patient suffering chronic kidney disease, harbors genes with a functional role in virulence and antimicrobial resistance [20], highlighting the need for subsequent studies to further characterize the virulence potential of this commensal as an opportunistic human pathogen. To expand on the characterization of *C. werkmanii* as an emerging opportunistic pathogen, the present study conducted comparative genomic analyses of *C. werkmanii* strains recovered from a major agricultural region producing horticultural products in Mexico, as well as from various geographical locations and sources, to identify relevant virulence and antimicrobial resistance genes. This research identified key determinants implicated in highly pathogenic and multidrug-resistant profiles in *C. werkmanii* strains and has provided fundamental information to enable the characterization of this emerging and opportunistic pathogen in humans.

## 2. Materials and Methods

### 2.1. Bacterial Strain Isolation and Growth Conditions

The examined *Citrobacter* strains, LANIIA-031 and LANIIA-032, were isolated from a field survey study conducted in major agricultural rivers in the state of Sinaloa in Northwestern Mexico [21]. Specifically, a size-exclusion ultrafiltration method was employed to efficiently concentrate large river water volumes, resulting in a final suspension of the targeted bacterial species in the retentive volume, as recently documented in [21]. The strains named LANIIA-031 and LANIIA-032 were recovered after subjecting the concentrated river water samples to an enrichment step in tryptic soy broth (Becton Dickinson Bioxon, Mexico City, Mexico) at 37 °C for 24 h, followed by subsequent growth on xylose lysine desoxycholate selective agar, as described in a previous study [21]. The strains were subsequently preserved at −80 °C in a 50/50 mixture of glycerol and tryptic soy broth for further genomic characterization.

### 2.2. Genome Sequencing and Annotation

To characterize the recovered *Citrobacter* strains from river water, the LANIIA-031 and LANIIA-032 strains were subjected to whole-genome sequencing. DNA extractions were performed using the DNeasy Blood and Tissue kit (QIAGEN, Mexico City, Mexico), following the manufacturer’s specifications, and the recovered genomic DNA was initially assessed and quantified with a NanoDrop 2000c spectrophotometer (Thermo Fisher Scientific, Waltham, MA, USA). For performing the sequencing reactions, genomic DNA from the *Citrobacter* strains was quantified with a Qubit™ 2.0 Fluorometer (Invitrogen, Carlsbad, CA, USA) and adjusted to a 0.2 ng/μL concentration. The genomic DNA libraries per strain were prepared using a final amount of 1 ng with the Nextera XT DNA Library Preparation Kit (Illumina Inc., San Diego, CA, USA). They were then sequenced using a MiSeq™ Reagent Kit v2 (300-cycle format) to obtain a 2 × 150 bp paired-end read output with a MiSeq™ System (Illumina, Inc.) at the Earlham Institute (Norwich Research Park, Norwich, United Kingdom), as in previous studies [22]. To initially assess the quality of the sequencing output prior to assembly, the raw sequence data were visualized and evaluated using the FastQC program [23]. The script wrapper Trim-Galore, version 0.6.4 [24], was employed for the removal of low-quality bases (value < 30) from the 3′-end of the reads, of adapter sequences, and of reads shorter than 80 bp in length, and the Clumpify tool, version 38.75 [25], was used to eliminate the duplicated sequencing reads. For the de novo read assembly, the A5-miseq pipeline, version 20160825 [26], was used with the re-scaffolding process to reduce the contig numbers, and the results were chosen according to the final number of contigs, N50, and genome size. To perform genome annotation, Prokka software, version 1.14.5 [27] and the RAST server [28] for the *Citrobacter* genus (NCBI:txid544) were used with a total of 30 genomic sequences, downloaded from the National Center for Biotechnology Information, detailing *C. werkmanii* strains from various geographical locations and sources.

To screen the contig assemblies for the examined *C. werkmanii* strains, the virulence finder database and the comprehensive antibiotic resistance database [29,30] for genes related to virulence and antimicrobial resistance traits, respectively, were employed in conjunction with the ABRicate tool, version 1.0.1, by selecting the parameter cutoffs of 90% coverage and 95% nucleotide identity [31]. The ABRicate results were then represented on a clustered heatmap, depicting the presence or absence of the virulence and antimicrobial resistance gene profiles, and the heatmap was constructed and edited with the Interactive Tree Of Life (iTOL) Annotation Editor, version 5 [32]. Genes encoding the proteins linked to the various bacterial secretion systems were detected by employing the program MacSyFinder, version 1, with the TXSScan model, version 1.1.1 [33,34], by using a maximal E-value of 0.001 as the statistical threshold of significance for the analysis of secretion systems.

### 2.3. Phylogenetic, Comparative Genomics, and Pangenome Analyses

The phylogenetic analysis of the *C. werkmanii* strains, LANIIA-031 and LANIIA-032, recovered from river water was performed using the concatenated sequences of the housekeeping genes, *fusA* (protein synthesis elongation factor-G), *leuS* (leucine tRNA synthetase), *pyrG* (CTP synthetase), and *rpoB* (β-subunit of RNA polymerase), as an established sequence-based method for the identification of species within the genus *Citrobacter* [35]. The sequence of *recN* (DNA repair) was also employed as another reliable marker for molecular species identification [36]. For the phylogenetic analyses, the genome data from the *Citrobacter* type strains (Appendix A) were also included in the phylogenetic analyses. Sequence concatenation and alignments were performed with Geneious software, version 9.1.8 (Biomatters, Auckland, New Zealand) and were imported into MEGA-X [37] for constructing the phylogeny trees, using the neighbor-joining method [38]. The phylogenetic tree was rooted at the midpoint, then the topology was validated by performing a bootstrap test for a total of 1000 replicates. The evolutionary distances were computed using the maximum composite likelihood method, with a uniform rate for substitution [39].

Based on the results of the phylogenetic trees, the genomic sequences from closely related *Citrobacter* species were subsequently selected for further comparison with the strains LANIIA-031 and LANIIA-032, based on documented methods for phylogenetic analysis using the complete genome data [40]. Digital DNA–DNA hybridization was performed with the server located at: https://www.dsmz.de/services/online-tools/genome-to-genome-distance-calculator-ggdc (accessed on 9 December 2022), using “suggest method 2” [41]. The average nucleotide identity values between genomes and coverage were determined using JSpecies, version 1.2.1 [42], and the reported average nucleotide identity values were adjusted by the genome coverage on the JSpecies web server. To initially examine genomic variability in the recovered strains of LANIIA-031 and LANIIA-032, the genome assemblies were submitted to the IslandViewer 4 server [43] to search for genomic islands. To further characterize the genomic islands, the genomes were examined for identifying phages using PHASTER [44]. Comparison of the genomes of the strains LANIIA-031 and LANIIA-032 was performed using progressiveMauve [45], and the diagram with the genomic islands was generated with Geneious software, version 9.1.8 (Biomatters).

To examine the complete genetic composition, using the available sequence data of the *C. werkmanii* strain collection for the phylogenetic clade grouping, a pangenome was constructed using a combination of Roary, version 3.11.2 [46], and Anvi’o, version 7 [47] platforms. Phylogenetic relationships based on the core genomes of *C. werkmanii* were generated with the HarvestTools suite, version 1.2.2, by employing the multi-aligner, Parsnp, version 1.7.4, and visual platform, Gingr, version 1.3, to obtain a multi-FASTA alignment [48]. The RAxML program, version 8 [49], with a general time-reversible model, was used to construct a core genome phylogenetic tree using a gamma distribution and 1000 bootstrap replicates. The iTOL program, version 5 [32] was used to visualize and edit the core phylogenetic tree. Additionally, GFF files were generated with the Prokka tool, version 1.14.5; these files were used as input for generating a gene alignment and for identifying the presence of core and accessory genes, based on a 90% identity with the Roary program, version 3.11.2. The pangenome was visualized with Phandango, version 1.3.0 [50], and some additional plots were generated using the script entitled “roary_plots.py”. Moreover, the Anvi’o platform, version 7, was used by selecting the pangenomics workflow system (available at https://merenlab.org/2016/11/08/pangenomics-v2/ and accessed on 18 November 2021) for constructing the pangenome of the *C. werkmanii* strains. The results obtained using the annotated genomic databases were subsequently visualized using the command entitled “Anvi-display-pan”. The gene calls were clustered into bins, based on the following criteria: (i) core genes present in 99–100% of the genomes; (ii) shell genes present in 15–99% of the genomes; and (iii) cloud genes present in less than 15% of the genomes [46,51]. Furthermore, the data in the gene cluster summary file from the Anvi’o workflow system was used to analyze the distribution of functional clusters of orthologous gene (COG) categories in the shell and cloud pangenome in the examined *C. werkmanii* strains.

### 2.4. Statistical Analyses

Statistical analyses were performed by conducting Fisher’s exact test, using the R Statistical Software (version 4.2.0; R Foundation for Statistical Computing, Vienna, Austria) [52]. Probability values (*p*-values) of lower than 0.05 were considered significant.

## 3. Results

### 3.1. Characterization of Citrobacter Strains from River Water

Phylogenetic analyses of *Citrobacter* reference strains (Appendix A), based on concatenation of the MLST genes *fusA*, *leuS*, *pyrG*, and *rpoB* (Figure 1a) and also of the housekeeping *recN* gene (Figure 1b), revealed that the river water strains LANIIA-031 and LANIIA-032 belong to the *C. werkmanii* species and were found to cluster together in a separate branch with the reference strain *C. werkmanii* FDAARGOS_364^T^ as the most closely related species. Based on the results of the phylogenetic trees (Figure 1), the genomic sequences of the *Citrobacter* river water strains, LANIIA-031 and LANIIA-032, were further examined by calculating the digital DNA–DNA hybridization, in comparison to the genome data from closely related *Citrobacter* reference strains (Appendix A). As shown in Table 1, the recorded values of the digital DNA–DNA hybridization analysis revealed that strains LANIIA-031 and LANIIA-032 had values of >90% when compared to the reference strain, *C. werkmanii* FDAARGOS_364^T^. In contrast, the digital DNA–DNA hybridization values of strains LANIIA-031 and LANIIA-032 when compared to other species were below 70%, which is the established threshold for bacterial species delineation [5,53]. As an additional in silico method for corroborating the speciation of the river water strains, LANIIA-031 and LANIIA-032, the genomes were compared via the average nucleotide identity method (Appendix A), and the results showed values above 90% for the LANIIA-031 and LANIIA-032 strains when compared to the reference strain, *C. werkmanii* FDAARGOS_364^T^. When compared to other *Citrobacter* species, values below 79% were observed; these findings were in agreement with the phylogenetic analysis and digital DNA–DNA hybridization, indicating that the river water strains of LANIIA-031 and LANIIA-032 are *C. werkmanii*.

### 3.2. Pangenome Analyses of C. werkmanii Strains from Diverse Sources and Locations

To obtain a better understanding of the virulence potential in *C. werkmanii*, a pangenome analysis was conducted by employing the Roary and Anvi’o workflows with an additional thirty publicly available genomic data points from *C. werkmanii* strains (Table 2), recovered from distinct geographical locations and sources. For this pangenome analysis, the resulting gene calls were clustered into bins based on: (i) core genes present in 99–100% of the genomes; (ii) shell genes present in 15–99% of the genomes; and (iii) cloud genes present in less than 15% of the genomes. A total of 11,680 gene clusters with 153,862 genes were detected to be specific for the *C. werkmanii* pangenome, which was classified as a core, shell, and cloud genome. As indicated by the outer colored ring in the pangenome diagram (Figure 2), the core genome in the examined *C. werkmanii* strains was the most abundant and consistently detected fraction (99–100% of the strains), with 3871 gene clusters and 126,303 gene calls. Moreover, 367 gene clusters and 6832 gene calls were detected for the shell genome (15–95% of the strains). Finally, the cloud genome (0–15% of the strains) accounted for 7442 gene clusters and 20,727 gene calls. The construction of a Heap’s Law chart showed that as new genomes were added to the analysis, the number of conserved genes (the core genome) decreased slightly (Figure 2, solid line), but the total number of genes increased considerably (Figure 2, dashed line). These observations were indicative of an open pangenome for *C. werkmanii* where the genomic content increased, with the number of additionally sequenced strains contributing to the species characterization.

Phylogenetic analysis of the *C. werkmanii* strains divided the pangenome of this species into five distinct clades (Figure 2 and Table 2). The phylogram, based on gene presence or absence, revealed that most examined *C. werkmanii* strains were clustered in two major clades, namely, clades IV and V. In particular, clade IV was composed of ten strains, including the two river-derived *C. werkmanii* strains, LANIIA-031 and LANIIA-032, from this study, as well as strain BF-6 from industrial water, strains Colony242 and Colony247 from food, and the environmental strains CVM45667, CVM45620, and CVM45672. A correlation analysis revealed that this clade was significantly associated with environmental sources (Fisher’s exact test *p*-value of < 0.01). Moreover, clade V, with the largest number of strains (Figure 2 and Table 2), a total of 16 strains, comprised *C. werkmanii* strains that had predominantly been recovered from clinical samples, associated with minor and severe human illnesses. The clinical strains included AK-8, ICR003007, CRE806, CRE1173, CRE346, 2580, CRK0001, and YDC667-1. Other isolation sources for the strains in clade V included chicken (strains JS97 and L38), sprouts (strain C5.1), and coastal water (strain CW_LB-887), and a significant correlation was identified for clade V strains with sampling sources from diverse geographical locations, excluding the United States (Fisher’s exact test *p*-value of <0.02).

Additionally, the functional COG categories associated with the *C. werkmanii* accessory genome are presented in Figure 2. When examining the entire pangenome, the average percentage of functional COG was 85%, and subsequent analysis of the gene clusters for the core, shell, and cloud genomes revealed the average percentage of the functional COGs to be 91.5%, 26.5%, and 57.8%, respectively. Carbohydrate metabolism and transport, cell wall structure, translation, transcription, signal transduction, and general metabolism (amino acids, lipids, and nucleotides) were detected as the principal orthologous groups of proteins during the analysis of the core genome. These orthologous protein groups in the core genome were associated with *C. werkmanii* traits that are needed for growth and survival, while the shell genome was found to be mainly composed of genes that are implicated in the secretion system, pilus assembly protein, and phage structural genes. When compared to the core genome, the analysis of the cloud and shell genomes showed significant variability.

By conducting a subsequent analysis with the Anvi’o software version 7, the presence of putative genes for strains belonging to each clade was identified and they were assigned to functional categories for the shell (Figure 3a) and cloud (Figure 3b) pangenomes. The cellular process and signaling (brown-yellow) category, including genes with a proposed function in kinase signal transduction, accounted for about 26% to 39% of the genes in the shell pangenome. Approximately 25% to 32% of the genes were assigned to this functional category in the cloud pangenome. Moreover, the information storage and processing (blue) category, including genes with a proposed function in transcription, showed the lowest percentage, ranging from 14% to 18% of the genes in the shell genome and 12% to 21% of the genes in the cloud pangenome. Interestingly, the most variability was observed for metabolic functions (pink-red), such as the transport of lipids, carbohydrates, amino acids, and nucleotides. The results for this category had the highest percentages of 34 to 38% of the genes, as detected for the human strains in clade I and the environmental clade IV for the cloud and shell genomes, respectively. In addition, a high percentage of uncharacterized genes (grey-black), ranging from 23–33%, were predominantly identified in the cloud genomes for strains belonging to all clades, and the cloud genomes for the human strains in clade II and clade V were identified by having prophages, transposons, and over 20% of the genes associated with the mobilome, corresponding to those genetic elements that can confer movement within and among the different bacterial genomes [54].

### 3.3. Virulence and Antimicrobial Resistance Gene Identification

To gain insights about specific virulence and resistance markers in *C. werkmanii*, the available genome data from a collection of strains recovered from various geographical locations and sources were analyzed (Table 2). Additional *Citrobacter* species that are implicated in human illness, the *C. koseri* strain ATCC BAA-895 and the *C. freundii* strain FDAARGOS_549, were added to the comparative analysis. A heatmap with a phylogenetic tree was constructed by indicating the presence or absence of targeted genes that are implicated in virulence and antimicrobial resistance (Figure 4).

The analysis revealed that virulence traits related to the flagellum, type I, and type Va secretion systems were detected consistently in all of the examined *Citrobacter* strains. Interestingly, the phylogenetic analysis revealed the presence of mandatory secretion system-specific genes implicated in the type II and type Vb secretion system for the strains in clade V (Figure 4, top panel), comprising the largest number of examined *C. werkmanii* strains that were predominantly recovered from the clinical samples. The virulence mapping analysis also revealed that more than 55% of the examined *C. werkmanii* strains were observed to harbor genes related to the type VI-class 1 secretion system. Additionally, distinct profiles were also observed when examining the presence of the adhesin curli fimbrae *csgF* gene, the *entS* enterobactin gene, and the siderophore *iroBCDN* operon among the strains belonging to the different clades (Figure 4, top panel). In contrast, the adhesin genes *csgB* and *csgE* and the iron uptake genes *entB*, *entE*, and *fepC* were present in all examined *C. werkmanii* strains. By using the VirulenceFinder computational tool, genes implicated in the type VI secretion system were identified for *C. werkmanii* strains belonging to all five clades. Although a larger number of virulence genes were detected for the clinically relevant species, *C. koseri*, a total of 16 virulence genes were commonly detected for the *C. werkmanii* strains recovered from clinical, food, and environmental sources at various geographical locations.

Multidrug resistance was observed in the examined *C. werkmanii* strains, and resistance to beta-lactamase, efflux pumps, peptide antibiotics, fluoroquinolone, and aminocoumarin were commonly observed in this study (Figure 4, bottom panel). With the exception of the strains FDAARGOS_616 and CRK0001, the remaining *C. werkmanii* strains that were clustering in clade V, which were predominantly from clinical sources, were found to have a larger number of genes associated with antimicrobial resistance in various drug classes when compared to the other examined *C. werkmanii* strains, as well as the clinical *C. freundii* FDAARGOS_549 and *C. koseri* ATCC BAA-895 reference strains (Figure 4, bottom panel). In particular, distinct antimicrobial gene profiles were detected among the *C. werkmanii* strains belonging to clade V for aminoglycoside {*aac*(3)-*lle*, *ant*(2″)-*la*, *ant*(3″)-*lla*, and *ant*(3″)-*la*}, beta-lactamases (bla_OXA-10_, bla_OXA-2_, bla_OXA-370,_ and bla_OXA-48_), fluoroquinolone (*qnrA1*, *qnrB11*, *qnrB12*, *qnrB17*, and *qnrB20*), phenicol (*catl*, *catll*, *cmlA1*, and *cmlA5*), trimethoprim (*dfrA1*, *dfrA12*, *dfrA14*, *dfrA17*, *dfrA19*, *dfrA23*, and *dfrA27*), macrolide (*mphA* and *mphE*), sulfonamide (*sul1*, *sul2*, and *sul3*) and tetracycline (*tetA*, *tetB*, *tetC*, *tetD*, and *tetM*). The current analysis revealed that *C. werkmanii* strains belonging to clade V that were recovered from clinical sources showed the most extensive antimicrobial resistance profiles. In particular, a significantly higher number of resistance genes was observed for *C. werkmanii*, which had a total of 27 genes, when compared to the clinically relevant *C. koseri* strain ATCC BAA-895, with a total of 21 genes. In summary, these findings indicated that the *C. werkmanii* strains harbor virulence determinant genes, conferring an ability regarding host adaptation and colonization as well as multidrug resistance to various high-priority agents, indicating significant potential for *C. werkmanii* as an emerging and opportunistic pathogen in humans.

## 4. Discussion

In the present study, a size-exclusion ultrafiltration method, followed by genome sequencing and phylogenomic analyses, revealed the isolation of novel strains of *C. werkmanii*, the strains LANIIA-031 and LANIIA-032, which were recovered from agricultural river water in an important region for the production of fresh produce in northwestern Mexico. The examined *C. werkmanii* strains, LANIIA-031 and LANIIA-032, were isolated from major agricultural river samples as part of a field survey study aimed at the isolation of *Salmonella enterica* [21]. Preliminary typing and biochemical standardized assays falsely identified the LANIIA-031 and LANIIA-032 strains as *Salmonella*. However, the use of whole-genome sequencing led to the accurate species identification of the LANIIA-031 and LANIIA-032 strains as *Citrobacter*. In previous studies, phylogenomic analyses have indicated that *Salmonella* and *Citrobacter* have similar environmental niches and a highly related evolutionary history [55]. The genomic similarity between *Salmonella* and *Citrobacter* poses a challenge in food safety since current traditional methods may lead to the inaccurate speciation of these foodborne pathogens. These findings indicate that the use of high-resolution sequencing-based methods focused on genomic characterization enables the accurate species identification of the recovered bacterial pathogens from diverse samples relevant to food safety, environmental monitoring, and diagnostics testing.

Pangenome analyses of a large collection of sequenced *C. werkmanii* strains from diverse sources and locations resulted in the grouping of the strains into five distinct clades. Based on the identification of a gene’s presence or absence, as well as single nucleotide polymorphisms, most of the strains clustered in clade IV were from environmental sources and those in clade V were predominantly from clinical sources. The pangenome analysis from this study showed that the examined *C. werkmanii* strains had genomic features and a similar number of genes that were associated with subsystem categories, a finding that was in agreement with previous reports characterizing the genomic structure for another environmental *C. werkmanii* isolate recovered from industrial spoilage samples [56]. The current analysis also revealed that *C. werkmanii* exhibited an open pangenome, and this finding was in agreement with the characteristics of other *Enterobacteriaceae* family members such as *S. enterica* and *Escherichia coli* [57,58]. The supporting evidence indicating an open pangenome for *C. werkmanii* would imply a genome in a species that is constantly exchanging genetic material and implies an ability to colonize multiple environments, as observed for other pathogenic and environmental bacterial species [59].

In agreement with previous studies examining the pangenome of other *Citrobacter* species [60], the core pangenome in *C. werkmanii* comprised the essential gene components associated with general metabolism, replication, cell wall structure, and transcription categories in *C. werkmanii* strains belonging to all clades for this opportunistic pathogen. Although small variations between the functional gene classification among the strains were observed, the functional gene category analysis revealed that the shell and cloud genomes showed the highest variability in terms of the number of genes for cellular metabolism, including carbohydrate lipid, nucleotide, and amino acid transport, as well as secondary metabolite biosynthesis. In particular, carbohydrate metabolism genes were detected as the main accessory features, resulting in additional gene copies in the shell and cloud genomes for *C. werkmanii* strains belonging to all clades. In particular, genes with an important role in galactose metabolism were identified that were coding for galactose mutarotase and beta-galactosidase, which are involved in substrate conversion and degradation [61]. Previous studies have shown that galactose is an important carbohydrate for the lipopolysaccharide composition of *Citrobacter* species [62,63].

Interestingly, the cloud genomes for the human strains in clades II and V had a large number of prophages, transposons, and mobilome-associated genes. The acquisition of mobile elements is considered an important evolutionary mechanism for bacterial pathogens to adapt and colonize many reservoirs, as well as to enhance the virulence potential [64,65,66]. Additionally, the presence of phage protein tail and capsid structural genes were found as part of both the shell and cloud genomes. Phages have been described as a common bacterial evolution mechanism involving horizontal gene transfer [54,66]. Moreover, the findings from this study indicated a high number of genes of unknown function or uncharacterized genes, which were also in the cloud and shell genomes. These results indicate the current challenges in the annotation of microbial genomes, supporting the need for improved computational pipelines to integrate uncharacterized genes in an evolutionary, biotechnological, and ecological framework as the volumes of genomic data continue to be deposited in public repositories for the improved characterization of a particular bacterial species [67,68].

An analysis of virulence determinants showed that *C. werkmanii* strains in this study harbor genes that are implicated in secretion, as with the flagellum, type I, and type Va secretion systems. To our knowledge, the present study is the first one to identify specific clades associated with genetic traits linked to the secretion system in this species. Moreover, genes related to the type II and type Vb secretion systems were found among the *C. werkmanii* strains that belonged to clade V, which were recovered predominantly from clinical sources. The type II secretion system is implicated in protein effector translocation across the outer membrane in pathogenic bacteria [69], while the type Vb secretion has the characteristic structure of a two-partner secretion pathway, with a role in the adhesion to receptors in mammalian cells [70]. Moreover, the identified type II secretion system genes in this study have a known role in the translocation of virulence factors, toxins, and enzymes across the cell’s outer membrane, as described for other human pathogens such as *E. coli*, *Vibrio cholerae*, *Chlamydia trachomatis*, and *Acinetobacter baumannii* [71]. Finally, genes of the type VI secretion system were identified in the *C. werkmanii* strains examined in this study. In a previous report, the type VI secretion system in *C. freundii* has been shown to contribute to adhesion and cytotoxicity in host cells [72]. Interestingly, the expression of the type VI secretion system by bacterial pathogens has been considered to be an antagonist mechanism in the colonization of the human gut by the resident microbiota [73], and the presence of the type VI secretion system could potentially provide *C. werkmanii* with a selective competitive advantage during the colonization of the human host.

Another important observation of the virulence categorization was that the curli fimbriae (Csg) and outer membrane protein A (OmpA) were detected in most of the examined *C. werkmanii* strains. In particular, curli has been well characterized in pathogenic *E. coli* and *S. enterica* and has a role in biofilm formation and in attachment to the extracellular matrix and colonization of the mammalian host cells [74]. Moreover, recent studies on the *C. werkmanii* strain BF-6 have demonstrated that OmpA plays an important role in the regulation of multiple virulence phenotypes, including biofilm formation on distinct surfaces, swimming motility, metal ion responses, and resistance to biocides [75]. Finally, the virulence typing analyses of *C. werkmanii* indicated the presence of the siderophore enterobactin genes for the acquisition of iron, an essential nutrient for bacteria growth and the increased virulence potential of *Citrobacter* [76,77].

Multidrug resistance to various antimicrobial agents was commonly observed among the examined *C. werkmanii* strains in this study. Several studies have proposed beta-lactam resistance as a common mechanism among *Citrobacter* species [78]. Resistance to the fluoroquinolone and aminocoumarin agents has previously been detected for other species of *Citrobacter* [60,77,79,80]. Interestingly, the present study demonstrated that the *C. werkmanii* strains that belong to clade V, which were predominantly from clinical sources, showed the most extensive antimicrobial resistance profiles. These observed multidrug resistances included various classes of antimicrobials, such as aminoglycoside, beta-lactamase, fluoroquinolone, phenicol, trimethoprim, macrolide, sulfonamide, and tetracycline. Due to the mortality rate associated with ineffective therapy in the treatment of *Citrobacter* infections [13], these observations highlight the importance of the resistome characterization of this opportunistic bacterial species. The global dissemination of multidrug-resistant enteric pathogens, including *Citrobacter*, has resulted in a high-priority action item under the One Health approach by integrating health solutions in various interfaces related to humans, animals, and the environment [81,82]. The spread of multidrug-resistant nosocomial infections to other niches is a critical issue that is in need of the improved identification and surveillance of bacterial virulence mechanisms and agent resistance [81]. In summary, this study demonstrated that the use of next-generation sequencing platforms in conjunction with bioinformatics allowed the identification in *C. werkmanii* of markers that are linked to virulence and antimicrobial resistance, indicating the potential of this species as an emerging and opportunistic pathogen in humans. 

## Figures and Tables

**Figure 1 microorganisms-11-02114-f001:**
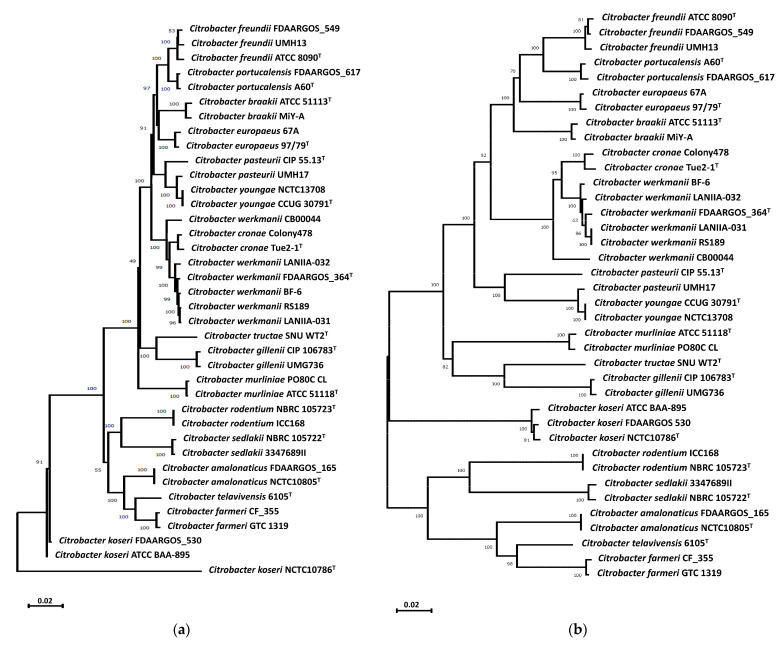
Phylogenetic relationships of the *Citrobacter* strains, LANIIA-031 and LANIIA-032, recovered from agricultural river water, to other *Citrobacter*-type strains. Maximum-likelihood phylogenetic trees were constructed, based on the concatenated *fusA*, *leuS*, *pyrG*, and *rpoB* genes (**a**) and the *recN* gene (**b**), by using the neighbor-joining method, rooted at the midpoint, and validated by performing a bootstrap test for a total of 1000 replicates. Bootstrap values (%) are indicated at the nodes, and the scale bar represents the expected number of nucleotide substitutions. The superscript “T” indicates a *Citrobacter* type strain.

**Figure 2 microorganisms-11-02114-f002:**
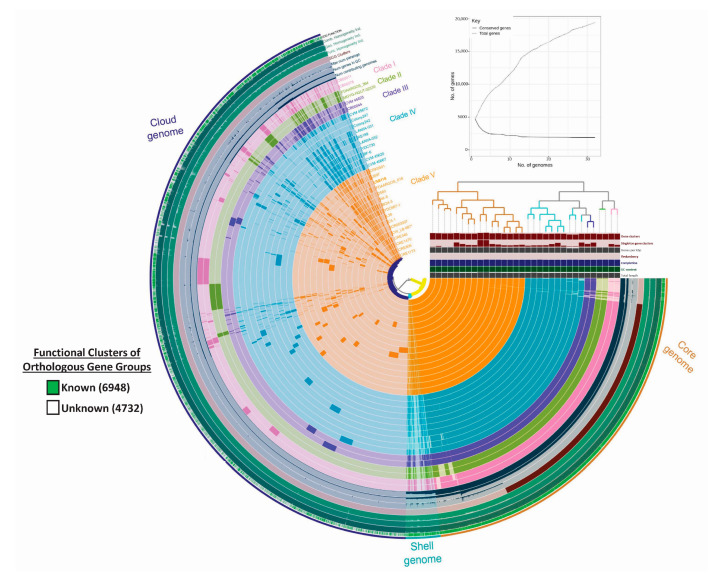
Pangenome analysis of the examined *Citrobacter werkmanii* strains. Pangenome analysis of over thirty *C. werkmanii* sequenced strains (Table 2) was constructed using Roary and Anvi’o platforms, and the resulting gene calls were clustered into bins, based on: (i) core genes present in 99–100% of the genomes; (ii) shell genes present in 15–99% of the genomes; and (iii) cloud genes present in less than 15% of the genomes. The construction of a phylogram, based on gene presence or absence in the examined *C. werkmanii* strains, resulted in the grouping of the strains into five distinct clades. Construction of a Heap´s Law chart (top right) showed that the number of conserved genes in the core genome decreased slightly (solid line) but the total number of genes increased considerably (dashed line), indicative of an open pangenome for *C. werkmanii*.

**Figure 3 microorganisms-11-02114-f003:**
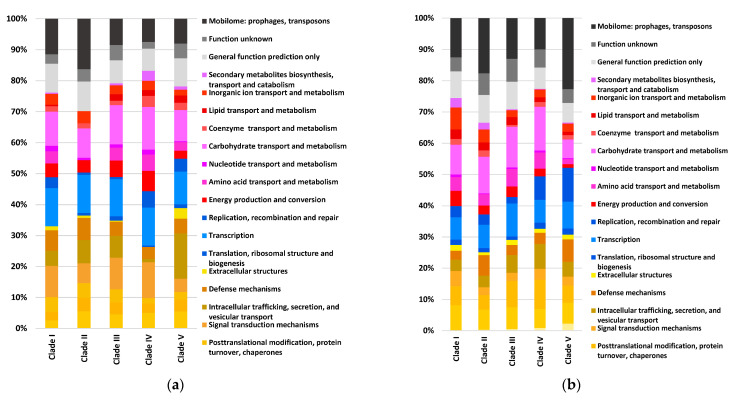
Functional COG categories derived from the *Citrobacter werkmanii* pangenome analysis. Over thirty *C. werkmanii* genomes (Table 2) were analyzed using Anvi’o software version 7, and putative genes were assigned to functional clusters of the orthologous gene categories for each clade, as described in Figure 2, for the shell genome (**a**) and cloud genome (**b**).

**Figure 4 microorganisms-11-02114-f004:**
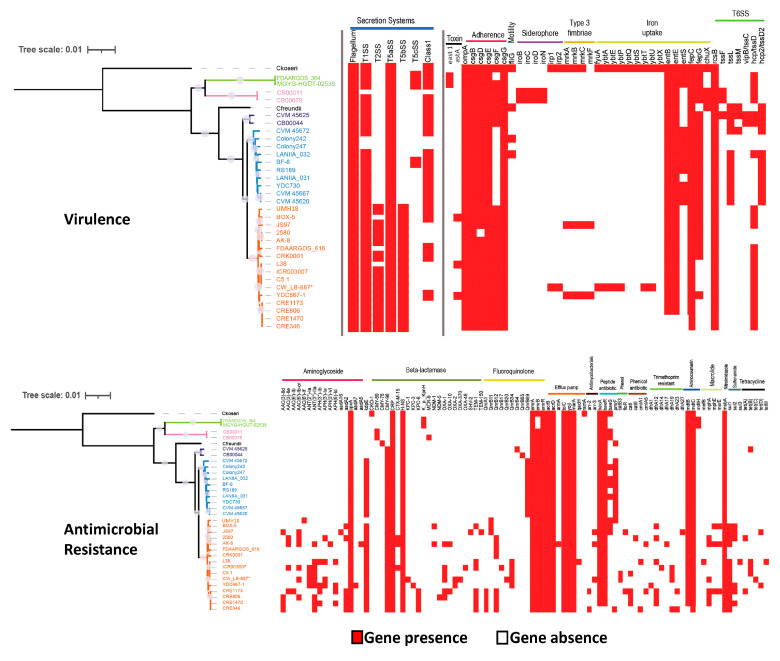
Virulence and antimicrobial resistance determinants in the examined *Citrobacter werkmanii* strains. A heatmap, indicating the presence or absence of virulence (**top panel**) and antimicrobial resistance genes (**bottom panel**), was constructed, coupled with a maximum-likelihood phylogenetic tree for the *C. werkmanii* strains belonging to the various clades, which are shown in distinct colors, as described in Figure 2, along with the clinical strains of *C. koseri* ATCC BAA-895 and *C. freundii* FDAARGOS_549. Gene presence in the heatmap is indicated by the red color and gene absence is indicated by the white color. Bootstrap values greater than 85% are shown with light blue circles over the phylogenetic tree nodes.

**Table 1 microorganisms-11-02114-t001:** Digital DNA–DNA hybridization of the *Citrobacter* strains examined in the present study.

*Citrobacter* Species ^1^	*C. werkmanii* LANIIA-032	*C. werkmanii* FDAARGOS_364 ^T^	*C. freundii* ATCC 8090 ^T^	*C. youngae* CCUG30791 ^T^	*C. pasteurii* CIP55.33^T^	*C. braakii* ATCC 51,113 ^T^	*C. europaeus* 97/79^T^	*C. portucalensis* A60 ^T^	*C. tructae* SNU WT2 ^T^	*C.cronae* Tue2-1^T^
*C. tructae* SNU WT2 ^T^										32.6
*C. portucalensis* A60 ^T^									33.0	42.4
*C. europaeus* 97/79 ^T^								50.3	32.9	42.9
*C. braakii* ATCC 51113 ^T^							52.9	48.5	33.1	42.7
*C. pasteurii* CIP 55.13 ^T^						38.8	38.5	38.4	32.9	36.2
*C. youngae* CCUG 30791 ^T^					59.6	39.0	39.0	39.4	32.9	36.5
*C. freundii* ATCC 8090^T^				39.8	35.2	45.0	45.2	52.4	29.8	37.6
*C. werkmanii* FDAARGOS_364 ^T^			37.8	36.4	36.1	42.7	43.4	42.4	32.4	70.0
*C. werkmanii* LANIIA-032		92.2	37.7	36.5	36.1	42.6	43.3	42.5	32.4	70.0
*C. werkmanii* LANIIA-031	92.3	92.2	37.9	36.5	36.2	42.7	43.3	42.5	32.4	70.0

^1^ The superscript “T” indicates a *Citrobacter*-type strain.

**Table 2 microorganisms-11-02114-t002:** List of *Citrobacter werkmanii* strains examined in the present study and their characteristics.

Clade ^1^	Strain ^2^	GenBank Accession Number	Isolation Source Description	Sample Source Type	Date ^3^	Country
Clade I	CB00011	GCA_016505505.1	Sputum	Human	2017	United States
	CB00078	GCA_016507625.1	Wound	Human	2018	United States
Clade II	FDAARGOS_364 ^T^	GCA_002386385.1	Stool	Human	2014	United States
	MGYG-HGUT-02535	GCA_902388105.1	Gut	Human	2019	United States
Clade III	CB00044	GCA_016505055.1	Not collected	Human	2017	United States
	CVM 45625	GCA_015942525.1	Unknown	Environmental	2019	United States
Clade IV	BF-6	GCA_002025225.1	Industrial water	Environmental	2012	China
	Colony242	GCA_016893825.1	Food	Food	2019	Thailand
	Colony247	GCA_016893645.1	Food	Food	2019	Thailand
	CVM 45620	GCA_015943365.1	Unknown	Environmental	2019	United States
	CVM 45667	GCA_015943485.1	Unknown	Environmental	2019	United States
	CVM 45672	GCA_015943405.1	Unknown	Environmental	2020	United States
	LANIIA-031	JAUJUK000000000	River water	Environmental	2018	Mexico
	LANIIA-032	JAUJUL000000000	River water	Environmental	2018	Mexico
	RS189	GCA_015958985.1	Not collected	Human	2017	United States
	YDC730	GCA_015958865.1	Pelvic abscess	Human	2015	United States
Clade V	2580	GCA_009907085.1	Urine	Human	2015	Nigeria
	AK-8	GCA_002114305.1	Human urine	Human	2014	India
	BOX-5	GCA_009856875.1	Hospital sink	Environmental	2016	France
	C5.1	GCA_008364715.1	Sprouts	Food	2015	Germany
	CRE1173	GCA_018106225.1	Pus	Human	2015	Malaysia
	CRE1470	GCA_018106165.1	Peritoneal fluid	Human	2016	Malaysia
	CRE346	GCA_018106145.1	Foot ulcer	Human	2015	Malaysia
	CRE806	GCA_018106185.1	Foot ulcer	Human	2014	Malaysia
	CRK0001	GCA_002185305.2	Blood	Human	2014	United States
	CW_LB-887	GCA_013303045.1	Coastal water	Environmental	2014	Brazil
	FDAARGOS_616	GCA_008693645.1	Clinical isolate	Human	NA	United States
	ICR003007	GCA_004146135.1	Hospital patient	Human	2017	France
	JS97	GCA_009821535.1	Chicken	Animal	NA	Unknown
	L38	GCA_013618825.1	Chicken liver	Animal	2019	Nigeria
	UMH18	GCA_003665555.1	Bacteremia	Human	2013	United States
	YDC667-1	GCA_013336965.	Lung tissue	Human	2014	United States

^1^ Clade grouping was based on whole genome analysis of *C. werkmanii* strains, as described in Section 2, Materials and Methods. ^2^ The superscript “T” indicates a *Citrobacter*-type strain. ^3^ Date refers to the sample collection date. NA refers to data that are not available.

## Data Availability

The complete genome sequences of the *C. werkmanii* strains of LANIIA-031 and LANIIA-032 can be found under the GenBank accession numbers JAUJUK000000000 and JAUJUL000000000, respectively, with the BioProject accession number PRJNA992637 in the National Center for Biotechnology Information (NCBI) BioProject database (https://www.ncbi.nlm.nih.gov/bioproject/ accessed on 15 July 2023).

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
