# Peer review of "Comparative Genomic Analyses of Virulence and Antimicrobial Resistance in Citrobacter werkmanii, an Emerging Opportunistic Pathogen"

_microorganisms, 2023, doi:10.3390/microorganisms11082114_

Round 1
Reviewer 1 Report
In the present study comparative genomic analysis of C. werkmanii strains from diverse geographical locations and sources was performed. The use of next-generation sequencing in conjunction with bioinformatics have demonstrated highly pathogenic and multidrug resistant genetic profiles in C. werkmanii strains, indicating a virulence potential for this emerging human pathogen. Although the manuscript is extensive, it is written clearly and comprehensibly. The topic is interesting and will attract the attention of the readers.
Author Response
MANUSCRIPT ID: microorganisms-2531567
REVIEWER 1
AUTHORS’ RESPONSES TO REVIEWER’S COMMENTS
1. REVIEWER’S COMMENT: In the present study comparative genomic analysis of C. werkmanii strains from diverse geographical locations and sources was performed. The use of next-generation sequencing in conjunction with bioinformatics have demonstrated highly pathogenic and multidrug resistant genetic profiles in C. werkmanii strains, indicating a virulence potential for this emerging human pathogen. Although the manuscript is extensive, it is written clearly and comprehensibly. The topic is interesting and will attract the attention of the readers.
AUTHORS’ REPLY: The Authors thank Reviewer 1 for the positive and constructive feedback.

Reviewer 2 Report
Comparative Genomic Analyses of Virulence and Antimicrobial Resistance in Citrobacter werkmanii, an Emerging Opportunistic Pathogen
Authors: José R. Aguirre-Sánchez, Beatriz Quiñones, José A. Ortiz-Muñoz, Rogelio Prieto-Alvarado, Inés F. Vega-López, Jaime Martínez-Urtaza, Bertram G. Lee, and Cristóbal Chaidez
The purpose of this study was to conducte comparative genomic analyses of C. werkmanii strains recovered from a major agricultural region for horticultural products in Mexico as well as from various geographical locations and sources to identify relevant virulence and antimicrobial resistance genes.
General comments:
Fascinating and interesting article. The strategy of this article is perfectly clear and very well executed.
Major comments:
- Chapter 2.3 (lines 127 to 142): why not also use a resistance gene (e.g. ampC) to build the phylogenetic tree? In fact, there is a wide range of ampC sequences that can be used to distinguish the different clades of Citrobacter freundii, for example. It would be interesting to discuss this point too
- it should be pointed out that it is possible to identify Citrobacter werkmanii : Citrobacter werkmanii can be accurately identified by MALDI-TOF MS.
- In the article published by Zhang et al (doi.org/10.3389/fmicb.2023.1056790), the authors point out that it is not possible to identify Citrobacter precisely using an ANI approach (“However, Citrobacter farmeri and Citrobacter werkmanii are exceptions because they can be accurately identified by MALDI-TOF MS, but not by rMLST and ANI (Table 1), indicating that different identification methods have different accuracy for different species. Therefore, supplementary methods, such as digital DNA-DNA hybridization computation and biochemical characterization are needed for identifying some species”). Would it be possible to re-discuss this finding?
Minor Comments:
- Line 55: please “freundii” in italics
- Line 56: please “koseri” in italics
- Line 442: please correct “baumanii” by “baumannii”
- Line 443: please “C. werkmanii” in italics
Author Response
MANUSCRIPT ID: microorganisms-2531567
REVIEWER 2
AUTHORS’ RESPONSES TO REVIEWER’S COMMENTS
1. REVIEWER’S COMMENT: The purpose of this study was to conduct comparative genomic analyses of C. werkmanii strains recovered from a major agricultural region for horticultural products in Mexico as well as from various geographical locations and sources to identify relevant virulence and antimicrobial resistance genes.
General comments:
Fascinating and interesting article. The strategy of this article is perfectly clear and very well executed.
AUTHORS’ REPLY: The Authors thank Reviewer 2 for the positive and constructive feedback.
2. REVIEWER’S COMMENT: Chapter 2.3 (lines 127 to 142): why not also use a resistance gene (e.g. ampC) to build the phylogenetic tree? In fact, there is a wide range of ampC sequences that can be used to distinguish the different clades of Citrobacter freundii, for example. It would be interesting to discuss this point too.
AUTHORS’ REPLY: In the present study, phylogenetic analyses of Citrobacter reference strains based on concatenation of the MLST genes fusA, leuS, pyrG and rpoB and also of the housekeeping recN gene revealed that the river water strains LANIIA-031 and LANIIA-032 belong to the species C. werkmanii (lines 186-1890. To acknowledge the suggestion by the Reviewer, C. werkmanii, C. freundii, and several other Citrobacter species can be identified by the ampC gene. However, the use of the ampC gene method cannot be employed by incorporating other Citrobacter species that lack a chromosomal ampC gene such as Citrobacter amalonaticus, Citrobacter farmeri, Citrobacter gillenii, Citrobacter koseri, Citrobacter rodentium, and Citrobacter sedlakii, as previously documented by Jacoby et al., 2009 (Clin Microbiol. Rev., Vol. 22: 161-182). The inclusion of the of the MLST genes fusA, leuS, pyrG and rpoB and also of the housekeeping recN gene enabled the extensive inclusion of Citrobact.er species (Figure 1).
3. REVIEWER’S COMMENT: It should be pointed out that it is possible to identify Citrobacter werkmanii : Citrobacter werkmanii can be accurately identified by MALDI-TOF MS.
AUTHORS’ REPLY: As indicated by the Reviewer, the authors acknowledge that proteomics can lead to the species identification in the genus Citrobacter; however, the scope of the present study was to employ sequence-based typing and genomic approaches for the characterization of virulence and antimicrobial resistance genes in C. werkmanii strains to further evaluate the pathogenic potential of this species. While C. werkmanii single colonies can be accurately identified by MALDI-TOF Mass Spectrometry (MS), previous studies have also shown that MALDI-TOF MS has a lower identification accuracy and can misidentify intraspecies such as C. portucalensis and C. freundii, which are generally considered the most prevalent Citrobacter species in the health clinics, as previously reported by Rödel et al., 2019 (Eur. J. Clin. Microbiol. Infect. Dis. 38, 581–591.).
4. REVIEWER’S COMMENT: In the article published by Zhang et al (doi.org/10.3389/fmicb.2023.1056790), the authors point out that it is not possible to identify Citrobacter precisely using an ANI approach (“However, Citrobacter farmeri and Citrobacter werkmanii are exceptions because they can be accurately identified by MALDI-TOF MS, but not by rMLST and ANI (Table 1), indicating that different identification methods have different accuracy for different species. Therefore, supplementary methods, such as digital DNA-DNA hybridization computation and biochemical characterization are needed for identifying some species”). Would it be possible to re-discuss this finding?
AUTHORS’ REPLY: The Authors thank the comments from the Reviewer. The goal of the present study was to employ genomics for the characterization of the pathogenic potential of C. werkmanii strains recovered from distinct sources and geographical locations. As described in lines 370-379, the preliminary typing and biochemical standardized assays falsely identified the river strains LANIIA-031 and LANIIA-032 as Salmonella due to the similarities between Salmonella and Citrobacter. In the present study, the use of sequence-based typing (Figure 1) and whole genome sequencing (Figure 2) conclusively led to the accurate species identification of strains LANIIA-031 and LANIIA-032 as Citrobacter werkmanii. As described in the manuscript in lines 191-200, the use of DNA-DNA hybridization corroborated the sequence-based typing that the river isolates LANIIA-031 and LANIIA-032 were C. werkmanii and the results from the average nucleotide identity method (Supplementary information) provided some additional evidence of the species identification of the river isolates.
5. REVIEWER’S MINOR COMMENTS:
Line 55: please “freundii” in italics
Line 56: please “koseri” in italics
Line 442: please correct “baumanii” by “baumannii”
Line 443: please “C. werkmanii” in italics
AUTHORS’ REPLY: Lines 55, 56, 442, 443: Changed the formatting of the species using italics and corrected the spelling to “baumannii.”